

# A surface model for water and energy balance in cold regions accounting for vapor diffusion

Enkhbayar Dandar[1,2,4], Maarten W.Saaltink[2,3], Jesús Carrera[1,3], Buyankhishig Nemer[4]
[1]Institute of Environmental Assessment and Water Research (IDAEA), CSIC, c/ Jordi Girona 18-26, Barcelona,08034, Spain
[2]GHS, Department of Civil and Environmental Engineering, Universitat Politecnica de Catalunya, UPC, Jordi Girona 1-3,
Barcelona, 08034, Spain
[3]Associated Unit: Hydrogeology Group (UPC-CSIC)
[4]Department of Geology and Hydrogeology, School of Geology and Mining Engineering, MUST, Baga Toiruu 46-114,
Ulaanbaatar, 14191,Mongolia
*Correspondence to*: Enkhbayar Dandar (denkhbayar@gmail.com)
**Abstract.** Computation of recharge in subarctic climate regions is complicated by phase change and permafrost, causing
conventional conceptual land surface models to be inaccurate. We conjecture that large vapor pressure gradients, driven by
the large temperature difference between the soil surface and the thawing permafrost active layer, may cause a significant
water and energy transfer during late spring and early summer. To analyze this conjecture, we develop a two-compartment
water and energy balance model that accounts for freezing and melting and includes vapor diffusion as a water and energy
transfer mechanism. It also accounts for the effect of slope orientation on radiation, which may be important for high latitude
mountain areas. We apply this model to weather data from the Terelj station (Mongolia). We find that vapor diffusion plays
an important quantitative role in the energy balance and a relevant qualitative role in the water balance. Except for snowmelt
and a few large precipitation events, most of the continuous recharge is driven by vapor diffusion fluxes. Large vapor fluxes
occur during spring and early summer, when surface temperatures are moderate, but the subsoil remains cold, creating large
downwards vapor pressure gradients. Temperature gradients reverse in fall and early winter, but the vapor diffusion fluxes
do not, because of the small vapor pressure differences at low temperature. The downwards latent heat flux associated to
vapor diffusion is essential for the thawing of the active layer. On a yearly basis, it is largely compensated by heat
conduction, which is much larger than in temperate regions and upwards on average. Furthermore, we find that total surface
runoff is small and concentrated at the beginning of spring due to snowmelt. Recharge is relatively high and delayed with
respect to snowmelt because a portion of it is associated to thawing at depth, which may occur much later.





## 1. Introduction

This work is motivated by the assessment of water resources in the Upper Tuul River basin, around Ulaanbaatar (Mongolia) and, in general, by subarctic continental climate regions, characterized by very low temperatures, low rainfall and, yet, sizable runoff. This causes such regions to fall very low in the Budyko curve (see, e.g., Figure C1 of Hanasaki et al., 2008). That is, total runoff is much larger than what would be expected in terms of potential evapotranspiration and rainfall.

We conjecture that increased runoff may be caused by condensation (deposition) of air moisture. Condensation and freezing may be especially significant during spring when air temperature and moisture increase, which drives water vapor to the cold soil. Later in spring, the soil surface warms up, which causes the active permafrost layer to start thawing while maintaining large temperature gradients, which should drive a significant water (and latent heat) flux down wards by vapor diffusion. This mechanism was proposed by Shvetzov (1978) and it is also mentioned by Gusev and Nanosova (2002), but has never been analyzed. Since moisture condensation data are not available in such regions, a quantitative analysis requires the use of mathematical models. Moreover, since phase changes and vapor diffusion are driven by energy availability and temperature differences, these models must consider water and energy balances.

Water and energy balances are the basic building blocks for any hydrological model. Many types of models are used for modeling phenomena such as recharge, evapotranspiration and water flow in soil. We can divide them into FE/FD (Finite Element/Finite Difference) models and integrated models. FE/FD models use more or less finely discretized grids for solving the partial differential equations governing multiphase non-isothermal flow in unsaturated soils. Examples are CODEBRIGHT (Olivella et al., 1996), COUPMODEL (Jansson and Moon, 2001), HYDRUS (Šimůnek et al., 2008), and SHAW (Flerchinger, 2017). Many of them also consider ice, heat exchange and vapor diffusion. They have been extensively used to analyze permafrost dynamics (e.g., McKenzie and Voss, 2013), but they require detailed input parameters, which are scarce for subarctic regions such as the Upper Tuul River basin, and are hard to interpret. Therefore, integrated models may be more appropriate for our case.

Integrated models calculate the basic terms of the water balance, such as stored soil water, evapotranspiration and surface runoff, over vertically integrated portions of the domain (representing canopy, soil surface and/or other parts of the soil). These models receive different names depending on the branch of hydrology (or sister sciences) of the author. At the watershed scale, they are called "water balance models" (Yates, 1996) and "large area hydrological models" (Arnold et al., 1998). When applied to even larger (up to global) scales, they are also called "global hydrological models" by hydrologists, who emphasize water balance, or "land surface models" by climatologists, who emphasize the energy balance as well (Haddeland et al., 2011). Well-known examples at the watershed scale, are WATBAL (Yates, 1996) and SWAT (e.g., Arnold et al., 1998; Hülsmann et al., 2015). In its original form, SWAT calculates snowmelt by means of a water balance of the snow cover and a so called temperature-index method that estimates the snow cover temperature from previous ones and air temperature. Some later versions of SWAT use energy balances of the snow cover instead of the temperature-index method (e.g., Fuka et al., 2012). SWAT has been recently extended by considering energy balances in the soil as well (Qi et al., 2016), but only considers conduction as a mechanism for heat transport in the soil. None of these integrated models





simulate vapor diffusion in the soil. For our purposes we need to take into account vapor diffusion flux in both the water and
the energy balance of the soil.
The aim of this work is to assess the importance of vapor diffusion in cold and semi-arid regions and, so, obtain a better
understanding of the hydrological processes in such regions. We do this by developing a hydrological scheme that consider
both water and energy balances and accounts for vapor diffusion as well as other processes that are relevant for subarctic
climates, including the effect of slope on radiation.
**2. Methodology**
**2.1. Model description**
Water and energy balances in land surface hydrological models are typically expressed on one or two layers. The top layer
usually extends to root depth, where (shallow rooted) plants can extract water and daily temperature fluctuations are
dampened. We formulate the balances over two layers (see Figure 1) because we are interested in water and energy
dynamics including seasonal fluctuations. The surface layer extends some 16 cm, so as to accommodate the roots of typical
grass in the Upper Tuul basin and to dampen daily temperature fluctuations. The subsoil (with length $L_{ss}$) layer extends some
150 cm and accommodates the "active" layer that freezes and thaws seasonally. The input and output terms for the water
balance of the surface layer include precipitation (as rain or snow), evapotranspiration (including both ice deposition and
sublimation), infiltration into the subsoil and vapor diffusion to or into the subsoil. Those for the subsoil are infiltration,
vapor diffusion from the surface and recharge to the aquifer. The energy balance considers solar radiation, latent and sensible
heat fluxes, heat conduction between the two layers and energy released due to phase changes. The model also takes into
account the slope of the surface. The above fluxes can be written as a function of meteorological data and two state
variables: mass of water (kg m$^{-2}$) and energy (J m$^{-2}$). Details of each mass balance term are given below.





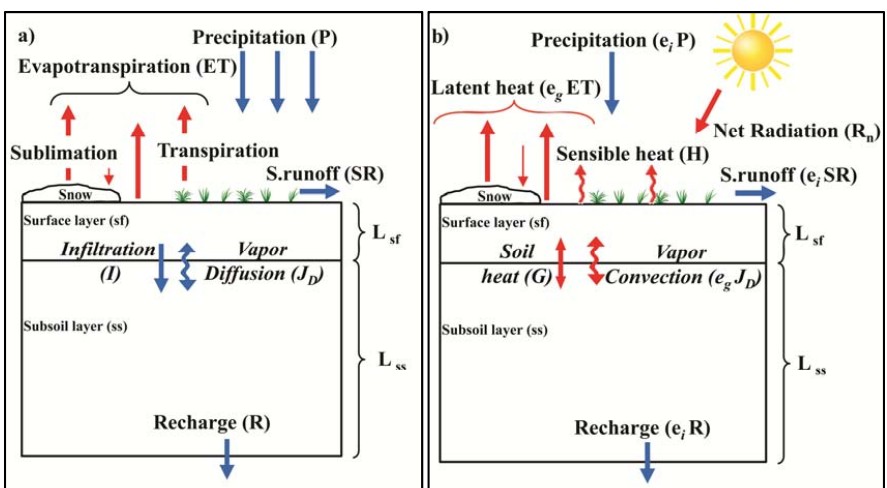

**Figure 1.** Schematic diagram of the water balance (a) and energy balance (b) models
**2.2. Water balance**
The water balance for the surface and the subsoil layers can be formulated as:

$$\frac{\partial m_{sf}}{\partial t} = P - ET_{sf} - I - SR - J_D \tag{1}$$

$$\frac{\partial m_{ss}}{\partial t} = I - ET_{ss} - R + J_D \tag{2}$$

where subscripts "sf" and "ss" refer to surface and subsoil layers, respectively, m is the mass of water (liquid or ice) (kg
m$^{-2}$), and fluxes (all in kg m$^{-2}$ s$^{-1}$) include precipitation (P), evapotranspiration (ET),infiltration (I, positive downwards),
surface runoff (SR), vapor diffusion (J$_D$, positive downwards), and recharge (R).
Evapotranspiration is commonly used to describe both evaporation and transpiration (Brutsaert, 1982), including
sublimation, i.e., the direct conversion of ice to water vapor (Zhang et al., 2004). ET is usually the most important term for
returning energy to the atmosphere. Traditional methods for estimating evapotranspiration can be divided into those that use
temperature (Hargreaves and Samani, 1985), radiation (Priestley and Taylor, 1972) and an aerodynamic approach
(McMahon et al., 2013 and Katul et al., 1992). We chose the latter, because in dry climates ET is controlled by water
availability rather than incoming radiation. The aerodynamic method is based on Dalton law, which establishes that ET is
proportionalto the difference of vapor pressures between air and soil. It is the basis of the method of Penman (1948) and has
been used to estimate evaporation from water surfaces (Rosenberg et al., 1983; Xu and Singh, 2002), or bare soil (Ripple et
al., 1970) and evapotranspiration from vegetated surfaces (Blad and Rosenberg, 1976). We assume that one part of the water
evaporates from the soil surface controlled by an aerodynamic resistance and another part evaporates through plant

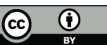



transpiration controlled by both aerodynamic and stomata resistance. Moreover, we distinguish between plant transpiration
from the soil surface and the subsoil. We express ET from soil surface and subsoil as:

$$ET_{sf} = \frac{M}{RT_{air}}\left[\frac{1}{r_a + r_s}\alpha\beta + \frac{1}{r_a}(1 - \beta)\right]f_{sf}[p_{v.sf} - p_{v.air}] \tag{3}$$

$$ET_{ss} = \frac{M}{RT_{air}}\ \left[\frac{1}{r_a + r_s}(1 - \alpha)\beta\right]f_{ss}[p_{v.ss} - p_{v.air}] \tag{4}$$

where M is the molar mass of water (0.018 kg mol[-1]), R is the gas constant (8.314 J mol[-1] K[-1]), $T_{air}$ is the air temperature (K),
$r_a$ and $r_s$ are the aerodynamic and stomata resistances, respectively (s m[-1]), α is the fraction of total transpiration from the
surface layer and βis the fraction of vegetation cover.When the soil is frozen, no transpirations is assumed andαβ are
given a zero value.Factor, f, represents the reduction of evaporation due to the lack of water in the surface and subsoil layers
(see below), $p_{v.sf}$ and $p_{v.ss}$ are the saturated vapor pressures(Pa) at the surface and subsoil layers and $p_{v.air}$ is the actual
vapor pressure in the atmosphere (Pa). The actual vapor pressure can be calculated from the relative humidity and saturated
vapor pressure ($p_{v.sat}$), which we computed with Murray's (1967) equation as a function of temperature.
The aerodynamic resistance,$r_a$, describes the resistance from the vegetation upward and involves friction from air flowing
over vegetative surfaces whereas the stomata surface resistance, $r_s$, describes the resistance of vapor flow through stomata,
total leaf area and evaporating soil surface (Shuttleworth, 1979). A general form for the aerodynamic resistance to
evapotranspiration (or sublimation) and sensible heat is (Evett et al., 2011):

$$r_a = \frac{1}{k^2 u_z}\left[\ln\left(\frac{z}{z_0}\right)\right]^2 \tag{5}$$

where z is height at which wind speed, temperature and relative humidity are measured (m), $z_0$ is the roughness length (m), k
is the von Karman's constant (k = 0.4) and $u_z$ is the wind speed (m s[-1]). The roughness length can vary over five orders of
magnitude (from 10[-5] m for very smooth water surfaces to several meters for forests and urban areas) and increases gradually
with increasing height of roughness elements (Arya, 2001).The stomata surface resistance can be calculated by(Allen et al.,

115    1998):

$$r_s = \frac{r_l}{0.5\ LAI} \tag{6}$$

where $r_l$ is the bulk stomata resistance of a well-illuminated leaf (s m[-1]). Monteith and Unsworth (1990) suggest that $r_l$=100
s m[-1] for grassland. LAI is the leaf area index (leaf area per unit soil surface area).
Factor, f, in equations (3) and (4) expresses the decreases of ET with water content in the surface and subsoil layers,
according to:





$$f_{sf/ss} = \begin{cases} 1 & \text{if } m_{sf/ss} > m_{sf/ss}^{fc} \\ \dfrac{m_{sf/ss} - m_{sf/ss}^{wp}}{m_{sf/ss}^{fc} - m_{sf/ss}^{wp}} & \text{if } m_{sf/ss} < m_{sf/ss}^{fc} \\ 0 & \text{otherwise} \end{cases} \qquad (7)$$

120 where superscripts fc and wp refer to field capacity and wilting point, respectively.

121 For the calculation of infiltration (I) we assume that it is limited by a maximum infiltration capacity, and that only liquid

122 water exceeding the field capacity can infiltrate. Then, for a time step $\Delta t$ the infiltration can be formulated as:

$$I^{k+1} = \max\left(0, \left[\min\left(\left(\frac{m_{sf}^{k+1} - m_{sf}^{wp}}{\Delta t} + P - ET_{sf} - J_D\right), I_{max}, m_{sf,l}\right)\right]\right) \qquad (8)$$

123 where $m_{sf}^{k+1}$ is the mass of water in the surface layer at a present time step and $m_{sf,l}$ is the liquid mass of water in the surface

124 layer (kg m$^{-2}$). The maximum infiltration ($I_{max}$) equals the saturated hydraulic conductivity ($K_{sat}$).

125 In a similar way we calculate surface runoff (SR) by assuming that only water exceeding the maximum water content in the

126 surface layer ($m_{sf}^{\phi}$) can runoff, where $m_{sf}^{\phi}$ (kg m$^{-2}$) equals the porosity ($\phi$) multiplied by water density and length of the soil

127 surface ($L_{sf}$). For the calculation of recharge (R) we assume that only liquid water exceeding the field capacity in the subsoil

128 layer can percolate to the aquifer.

129 Vapor diffusion using Fick's Law, is written as:

$$J_D = \frac{M}{RT_{sf}} \frac{D}{L_{sf}} \frac{m_{sf}^{\phi} - m_{sf}}{m_{sf}^{\phi}} \left[p_{v.sf} - p_{v.ss}\right] \qquad (9)$$

130 where D is the diffusion coefficient (m$^2$ s$^{-1}$), $T_{sf}$ is the temperature of the surface layer (K), $L_{sf}$ is the surface layer's length

131 (m). Note that we use $L_{sf}$ as length between the two layers rather than $(L_{sf} + L_{ss})/2$ because temperature gradients, which

132 control vapor pressure, are expected to be largest near the soil surface. The value to be adopted for the diffusion coefficient

133 deserves some discussion. The diffusion coefficient of water vapor in air is $0.3 \cdot 10^{-4}$ m$^2$ s$^{-1}$ (Cussler, 1997). This value should

134 be reduced due to reduced open area and tortuosity in a porous medium. However, for reasons that are subject to debate (Ho

135 and Webb, 2006), vapor diffusion is enhanced (Cass et al., 1984). Gran et al., (2011b) used values above $2 \cdot 10^{-4}$ m$^2$ s$^{-1}$. Given

136 these uncertainties we have adopted $10^{-4}$ m$^2$ s$^{-1}$ as base value, and then analyzed the sensitivity of the model to this

137 parameter.



140 **2.3. Energy balance**

141 Energy balance for the two layers is written as:


$$\frac{\partial U_{sf}}{\partial t} = e_{i/l}P - e_g\,ET_{sf} - (e_g - e_l)ET_{ss} - e_lI - e_lSR + R_n - H - G - e_gJ_D \qquad (10)$$

$$\frac{\partial U_{ss}}{\partial t} = e_lI - e_l\,ET_{ss} - e_lR + G + e_gJ_D \qquad (11)$$

where U is the total energy of each layer (J m$^{-2}$), $R_n$ is the net radiation (J m$^{-2}$ s$^{-1}$), H is the sensible heat flux (J m$^{-2}$ s$^{-1}$) and G
is the heat conduction (J m$^{-2}$ s$^{-1}$) between surface and subsoil. Advective heat fluxes include the fluxes of water, ice (snow)
or vapor (P, ET, SR, I, J, R) multiplied by the corresponding internal energies of liquid water, ice or vapor ($e_l$, $e_i$, $e_g$ (J
kg$^{-1}$)), which are linear functions of temperature. Although a distinction is made between evapotranspiration from the
surface and the subsoil (ET$_{sf}$ and ET$_{ss}$), equations (10) and (11) assume that the heat loss occurs only at the surface because
the actual phase change takes place in the plant, which is part of the surface layer.
The total energy ($U$) is a sum of energy for liquid water, ice and solid (Figure 2):

$$U = me = m_le_l + m_ie_i + m_se_s \qquad (12)$$

where subscript $s$ presents the solid part of the soil layers. During melting or freezing, the temperature is fixed at the melting
temperature of 0°C while at higher temperature all water is liquid ($m = m_l$) and at lower temperature all water is ice
($m = m_i$). Therefore, we can calculate the temperature and mass of ice and water as a function of $U$ and $m$ as:

$$T = \frac{U + m\Lambda_{melt}}{mc_i + m_sc_s} \quad \text{and} \quad m = m_i \qquad \text{if} \quad -m\Lambda_{melt} > U \qquad (13)$$

$$T = 0 \quad \text{and} \; m_i = -\frac{U}{\Lambda_{melt}}; \; m_l = m - m_i \qquad \text{if} \quad -m\Lambda_{melt} < U < 0 \qquad (14)$$

$$T = \frac{U}{mc_l + m_sc_s} \quad \text{and} \; m = m_l \qquad \text{if} \qquad U > 0 \qquad (15)$$

where $\Lambda_{melt}$ is the melting heat (3.34·10$^5$ J kg$^{-1}$), $c_l$ is specific heat of water (4184 J kg$^{-1}$ °C$^{-1}$), $c_i$ is specific heat of ice (2092
J kg$^{-1}$ °C$^{-1}$) and $c_s$ is specific heat of soil (843 J kg$^{-1}$ °C$^{-1}$). Equations (13) to (14) are also illustrated by Figure 2.



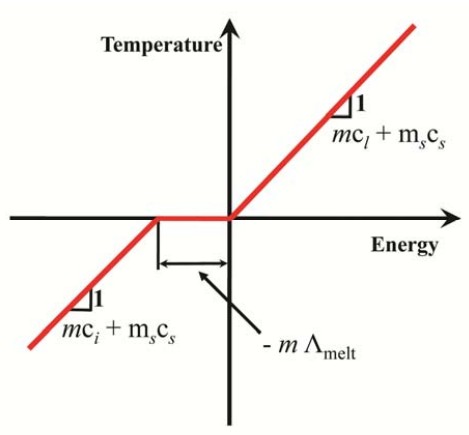

**Figure 2.** Temperature versus total energy

*Radiation*
Radiation is the main energy input for land surface models. Hence, it is not surprising that it has received a lot of attention.
We follow a somewhat modified version of the approaches of Tian el al., (2001) and Allen et al., (2006) and divide radiation
between shortwave, received from the sun, and longwave radiation, emitted by the Earth and the atmosphere. Net radiation
($R_n$) is the sum of the net shortwave radiation ($R_{ns}$) and the net longwave radiation ($R_{nl}$):

$$R_n = R_{ns} + R_{nl} = (1 - A)R_S + R_{L.up} + R_{L.down} \tag{16}$$

where A is the albedo, fraction of solar radiation ($R_S$) that is reflected by the surface. The albedo depends on surface types
(small for vegetated surface and high values for snow). In our model the albedo was determined from data on snow depth.
However, this could easily be changed (e.g., to a dependence on mass of ice in the subsoil), when no snow depth data are
available. The net longwave radiation is equal to the received (downward, $R_{L.down}$) minus the emitted ($R_{L.up}$) radiation.
*Solar radiation on a horizontal and on an inclined surface*
The solar radiation on a horizontal surface ($R_{S,hor}$) is measured or can be calculated from the atmospheric transmissivity
($\tau_a$), which is the fraction of extraterrestrial solar radiation (easy to compute, see Appendix A) that makes it to the land
surface. Atmospheric transmissivities are highly sensitive to cloudiness and moisture content. It can be estimated from the
relative sunshine hours (Allen et al.,1998) or through the method of Hargreaves and Allen (2003):

$$\tau_a = K_H\sqrt{T_{air}^{max} - T_{air}^{min}} \tag{17}$$

where $T_{air}^{max}$ and $T_{air}^{min}$ are the daily maximum and minimum air temperature and $K_H$ is an empirical constant. Allen et al.,
(1998) recommends $K_H = 0.16$ for interior and $K_H = 0.19$ for coastal regions.





Shortwave radiation may reach the land surface directly from the sun, reflected by the surrounding or scattered by the
atmosphere (diffuse solar radiation). The distinction is relevant for inclined surfaces in the shade, which only receive
reflected and diffuse solar radiation. Numerous relations can be found in the literature (Noorian et al., 2008) to estimate the
fraction of diffuse radiation over the total solar radiation ($f_{dif}$). We adopted the one of Boland et al., (2008), which is simple
and statistically sound:

$$f_{dif} = \frac{1}{1 + \exp(8.6\tau_a - 5)} \tag{18}$$

Solar radiation on an inclined surface can be calculated from the solar radiation on a horizontal surface by means of the
following expression (e.g., Tian et al., 2001):

$$R_S = R_{S,hor}\left[(1 - f_{dif})\frac{\max(\mathbf{p}^T\mathbf{s}, 0)}{s_{up}} + f_{dif}f_{sv} + A(1 - f_{sv})\right] \tag{19}$$

where $f_{sv}$ is the sky view factor (see equation A14 in the Appendix). The first term represents the direct solar radiation, the
second term the diffuse radiation and the third term the solar radiation reflected from the surroundings. Definitions of $\mathbf{p}^T\mathbf{s}$
and $s_{up}$ are shown in the Appendix (equations A1 and A2). To perform daily energy balances, we calculate the daily
averaged solar radiation on an inclined surface, assuming the atmospheric transmissivity ($\tau_a$) to be constant during the day, as
follows:

$$R_S = \left[(1 - f_{dif})\frac{\int_{-\omega_{ss}}^{\omega_{ss}}\max(\mathbf{p}^T\mathbf{s}, 0)\,dt}{\int_{-\omega_{ss}}^{\omega_{ss}}s_{up}\,dt} + f_{dif}f_{sv} + A(1 - f_{sv})\right]R_{S,hor} \tag{20}$$

where $\omega_{ss}$ is the sunset angle, the integrals of $s_{up}$ and $\max(\mathbf{p}^T\mathbf{s}, 0)$ are given by equations A6 and A13 of the Appendix.
***Longwave radiation***
Longwave radiation appears simple, but actual parameterization is hard (Herrero and Polo, 2012; Zabel et al., 2012). Upward
longwave radiation is calculated from Stefan-Boltzmann law:

$$R_{L.up} = -\varepsilon_s\sigma T_{sf}^4 \tag{21}$$

where $\varepsilon_s$ is the surface emissivity, $\sigma$ is the Stefan-Boltzmann constant ($5.7\cdot10^{-8}$ J s$^{-1}$ m$^{-2}$ K$^{-4}$) and $T_{sf}$ is the surface temperature
(K). The soil surface emissivity is usually close to 1 (Saito and Šimůnek, 2009). However, small changes in $\varepsilon_s$ may cause an
imbalance between upwards and downwards longwave radiation balance, thus having a large effect on net radiation. We
adopted a constant value of 0.94, but also tested 0.99 for sensitivity analysis purposes.
The Earth's surface also receives longwave radiation emitted by the atmosphere and surrounding surfaces. It can be
calculated from the same law:

$$R_{L.down} = f_{sv}\varepsilon_a\sigma T_{air}^4 - (1 - f_{sv})R_{L.up} \tag{22}$$




where $\varepsilon_a$ is the emissivity of the atmosphere and $T_{air}$ the absolute temperature of the atmosphere. Note that a fraction equal
to the sky view factor ($f_{sv}$) originates from the atmosphere and another part ($1-f_{sv}$) from the surroundings.
Clear sky emissivity is obtained from the empirical expression of Brutsaert (1975):

$$\varepsilon_{air.cs} = 1.24 (\frac{p_{v.air}}{T_{air}})^{1/7} \tag{23}$$

The cloudy sky emissivity ($\varepsilon_a$) is obtained from $\varepsilon_{air.cs}$ using the expression that Sicart et al., (2006) derived empirically for a
subarctic continental climate in Yukon (Canada):

$$\varepsilon_a = \varepsilon_{air.cs}(1 + 0.44h_r - 0.18\tau_a) \tag{24}$$

where $h_r$ is relative humidity.
*Sensible heat*
The sensible heat flux is calculated using the aerodynamic resistance ($r_a$) and soil surface resistance ($r_{sf}$):

$$H = \frac{\rho_a c_a}{r_a + r_{sf}}(T_{sf} - T_{air}) \tag{25}$$

where $\rho_a$ is the air density (1.22 kg m$^{-3}$) and $c_a$ is the specific heat of air (1013 J kg$^{-1}$ K$^{-1}$). The soil surface resistance, $r_{sf}$, is
calculated from the thermal conductivity of the soil:

$$r_{sf} = \frac{0.5 L_{sf} \rho_a c_a}{\lambda} \tag{26}$$

where $\lambda$ is the thermal conductivity of the soil.
*Heat conduction*
Heat conduction of soil can be calculated from Fourier's Law as:

$$G = \frac{\lambda}{(0.5\,L_{sf} + 0.5\,L_{ss})}(T_{sf} - T_{ss}) \tag{27}$$

where $\lambda$ is the soil thermal conductivity ($\lambda = \lambda_l^\phi \lambda_s^{1-\phi}$), where $\phi$ is the porosity, $\lambda_s$ is the thermal conductivity of the solid
particles, and $\lambda_l$, is the thermal conductivity of ice, $\lambda_i$, when water is frozen or that of liquid water, $\lambda_l$, otherwise (Côté and
Konrad, 2005).
**2.4. Numerical solution and implementation**
We solve the water and energy balance equations (Eqs. (1),(2), (10) and (11), respectively) using a semi-implicit finite
differences scheme with a time step of one day. The term semi-implicit means that all variables are treated explicitly (i.e.,
using the values from the previous day), with two exceptions to ensure stability. First, vapor pressures in equations (3), (4)





and (9) are linearized and treated implicitly (that is, $p_{v.sf}^{k+1} - p_{v.air} = (T_{sf}^{k+1} - T_{air})\Delta + (1 - h_r)p_{v.sat}$, where $\Delta$ is the slope
of the saturation vapor pressure curve (Pa °C⁻¹)). This type of linearization is frequent (e.g., Penman, 1948) because vapor
pressure is highly sensitive to temperature and treating it explicitly may cause instability. Second, we perform a preliminary
water balance to approximate the water available for evaporation, this is followed by the energy balance which yields not
only energy and temperature, but also actual evaporation, that is used for the final water balance.
The algorithm was implemented in a spreadsheet that is available for the cases discussed below at
http://h2ogeo.upc.edu/es/investigacion-hidrologia-subterrania/software.
All terms but the vapor diffusion, in the water (section 2.2) and energy (section 2.3) balances have been extensively tested
(see introduction). We test the validity of our formulation in the Appendix B.

### 2.5. Data of meteorological stations and parameters

The model was tested using meteorological data from 2000 through 2004 of the Terelj station (elevation 1540 m, 47.98N,
107.45E), located in northern Mongolia some 40 km east of Ulaanbaatar. This station records daily meteorological data
(maximum and minimum T, precipitation, snow depth, wind and relative humidity) provided by the Institute of Meteorology
and Hydrology of Mongolia. The area is mountainous with grassland and forest of Larix and Pinus. Forests dominate the
north face of mountains while grassland dominates the south face of mountains and flat areas (Dulamsuren et al., 2008;
Ishikawa et al., 2005). The region contains discontinuous and sparsely insular permafrost (Gravis et al., 1972; Sharkhuu,
2003; Jambaljav et al., 2008).
The average daily maximum and minimum air temperature is 5.06°C and -11.5°C, respectively. Mean air temperature
averaged -3.2°C for the studied period. Annual precipitation averaged 334 mm/year, with 80% falling between June and
September. Snow usually falls between mid-October and mid-April, with a maximum thickness of 31 cm. The average wind
speed is 1.5 m/s and average relative humidity 70.12%.
We used parameters from the literature (Table 1) to define a base model that assumes that the surface is horizontal and
covered by grass. Jambaljav et al., (2008) noted that the north and south facing slopes of mountains in the Terelj area are
about 10-40°.Therefore, for the sensitivity analysis, we considered north and south faces with a slope of 20 degrees.
Tuvshinjargal et al., (2004) used an albedo of 0.21 for grass meadow to calculate the surface energy balance. There are no
other data for albedo from this area, especially snow albedo.  The albedo (A) was taken from Oke (1987) as 0.6 during
periods with snow cover and 0.23 for grass and soil surface. Most surfaces have emissivities larger than 0.9 (Arya, 2001).
So, we used $\varepsilon_s$=0.94 for the base model. We assumed that all the roots of grass are in the surface layer ($L_{sf}$) which means the
evapotranspiration only occurs from the surface layer. Thus, we used $\alpha$ equal to 1.
The surface roughness length ($z_0$) is defined by surface types such as soil, vegetation and snow. We used $z_0$=0.04 for grass
and $z_0$=0.002 for snow surface (see figure 10.5 of Arya, 2001). The leaf area index (LAI) is defined by vegetation types.
Asner et al., (2003) give LAI of 2.1 m²m⁻² for grass. According to the National Soil Atlas of Mongolia (1981), the soil of the
study area belongs to the Gleysols-umbrisoland Cryosols-leptic type. The soil texture is mainly a silt-clay-loam. The wilting
point ($\theta^{wp}$), field capacity ($\theta^{fc}$), porosity ($\phi$) and saturated hydraulic conductivity ($K_{sat}$) are 0.11, 0.342, 0.365 and $4.2 \cdot 10^{-7}$ m




s$^{-1}$, respectively, and were obtained from Schroeder et al., (1994). Length of the surface layer (L$_{sf}$) and subsoil layer are of
0.16 m and 1.5 m, respectively. Thermal conductivities were obtained from Bristow (2002).
For the sensitivity runs we changed eight parameters: dip of the surface (θ), roughness length (z$_0$), soil emissivity (ε$_s$),
saturated conductivity (K$_{sat}$), wilting point (θ$^{wp}$), vapor diffusion coefficient (D), surface length (L$_{sf}$) and subsoil length (L$_{ss}$).

**Table1.** Parameter and values for Base model

| Parameters | Value | Units | Reference |
|---|---|---|---|
| Slope (θ) | 0 | degree | |
| Latitude (φ) | 48 | degree | |
| Albedo (A) | 0.23   for grass and soil<br>0.6     for snow | - | Oke, 1987 |
| Soil emissivity (ε$_s$) | 0.94 | - | Arya, 2001 |
| Fraction of transpiration (α) | 1 | - | |
| Vegetation cover (β) | 0.6 | - | Dulamsuren et al., 2008 |
| Leaf area index (LAI) | 2.1      for grass | m$^2$m$^{-2}$ | Asner et al., 2003 |
| Surface roughness length (z$_0$) | 0.04   for grass<br>0.002 for snow | m | Arya, 2001 |
| Diffusion coefficient (D) | 10$^{-4}$ | m$^2$s$^{-1}$ | |
| Field capacity (θ$^{fc}$) | 0.342 | m$^3$m$^{-3}$ | Schroeder et al, 1994 |
| Wilting point (θ$^{wp}$) | 0.11 | m$^3$m$^{-3}$ | Schroeder et al, 1994 |
| Porosity (φ) | 0.365 | - | Schroeder et al, 1994 |
| Saturated hydraulic conductivity (K$_{sat}$) | 4.2·10$^{-7}$ | m s$^{-1}$ | Schroeder et al, 1994 |
| Surface depth (L$_{sf}$) | 0.16 | m | |
| Subsoil depth (L$_{ss}$) | 1.5 | m | |
| Thermal conductivity (λ) | 2.9      for soil<br>0.57    for water<br>2.2      for ice | J s$^{-1}$m$^{-1}$K$^{-1}$ | Bristow (2002) |

**3. Results**
Results are summarized in Figure 3, which displays the evolution of the water fluxes (precipitation, evapotranspiration,
recharge, surface runoff and vapor diffusion), water contents, heat fluxes (net radiation, latent heat, sensible heat, vapor
convection and heat conduction) and temperature (air, surface and subsoil) of the base model during the last two years. Table
2 and 3 summarize the balances averaged over the 5 years for the base model and for the sensitivity.
Direct surface runoff is very small. Figure 3.b only shows some surface runoff at the beginning of April 2003 during
snowmelt. No surface runoff occurred during the snowmelt of 2004 probably because the accumulated snow on the surface
was small that year. The sensitivity analysis suggests that the limiting factor is the infiltration capacity. When saturated
hydraulic conductivity is reduced by a factor of 10, direct surface runoff increases dramatically and infiltration reduces. The
reduced infiltration also implies increased ET, so that the overall runoff (SR plus recharge) is also reduced. The small
surface runoff is consistent with the lack of very intense rainfall events and with the absence of indications of surface
erosion.



Infiltration and recharge are also relatively low. Infiltration occurs mainly after heavy rainfall events and it is not shown in
figure 3 because similar pattern as recharge. Most infiltration transforms into recharge because, in the absence of deep rooted
plants, the subsoil is always close to field capacity. Recharge from rainfall infiltration can vary a lot from year to year, due to
the irregular occurrence of heavy rainfall events. However, a significant amount of recharge occurs throughout the spring
and summer driven by vapor diffusion into the subsoil. While the rate is small (it can hardly be seen in Figure 3a, so we
zoom it in Figure 4), it occurs throughout the late spring and summer, after the subsoil has started to thaw. Overall, it is
about half of recharge from direct rainfall infiltration, but much more regular (it occurs every year) and quite robust, in that it
displays little sensitivity to model results(See Table 2).
Vapor diffusion between the surface and subsoil layers is positive (downwards) during spring and early summer, because
then the temperature and, therefore, vapor pressure is higher in the surface than in the subsoil. The flux fluctuates during late
winter, when the subsoil has started to warm, so that vapor diffuses upwards during cold days and downwards during warm
days. Diffusion is consistently upwards during autumn and winter, but the rate is very low because the saturated vapor
pressure is low and changes little with temperature (that is, $dp_{v,sat}/dT$ is small) at low temperatures. Therefore, there is a net
downward vapor flux. Its amount is not very large, but as mentioned above, it is what drives recharge during spring and early
summer.



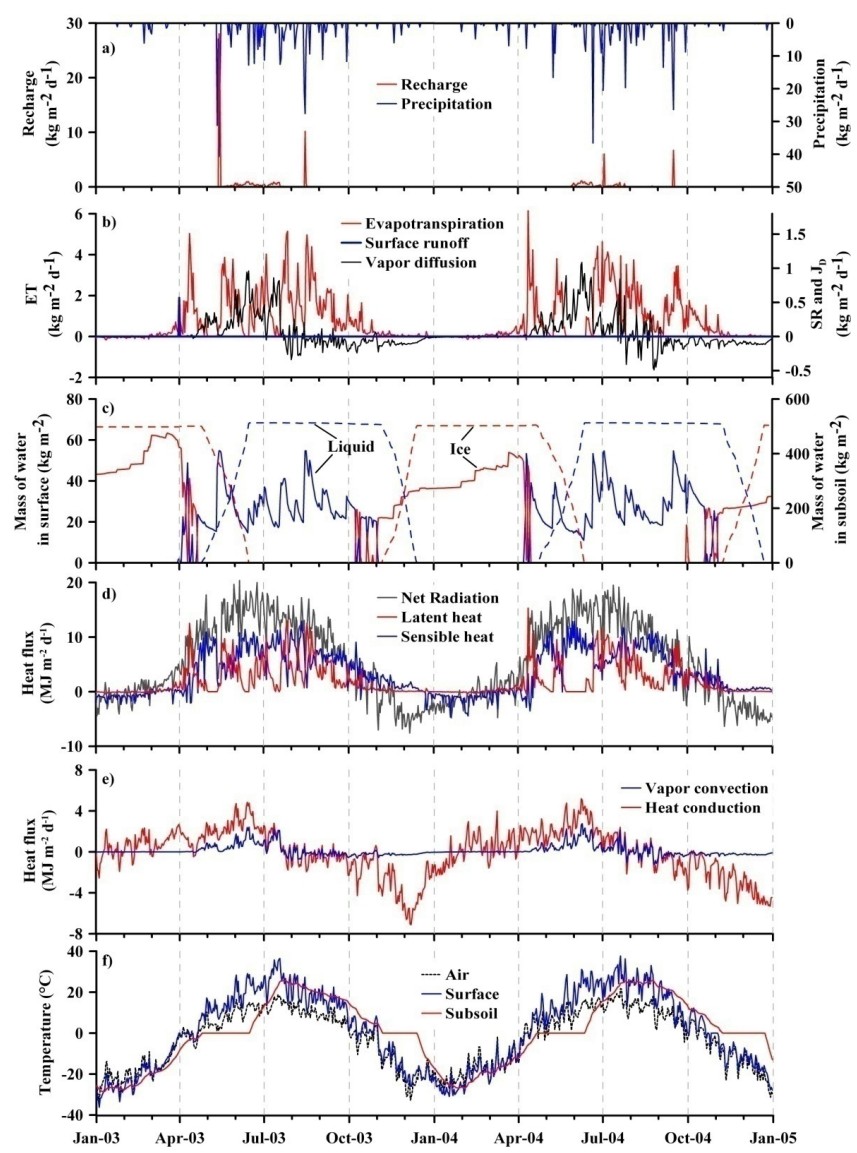

**Figure 3.** Daily evolution of (a) precipitation and recharge, (b) other water fluxes, (c) water content at surface (solid line) and subsoil (dashed line), (d, e) heat fluxes, and (f) temperatures of the base model



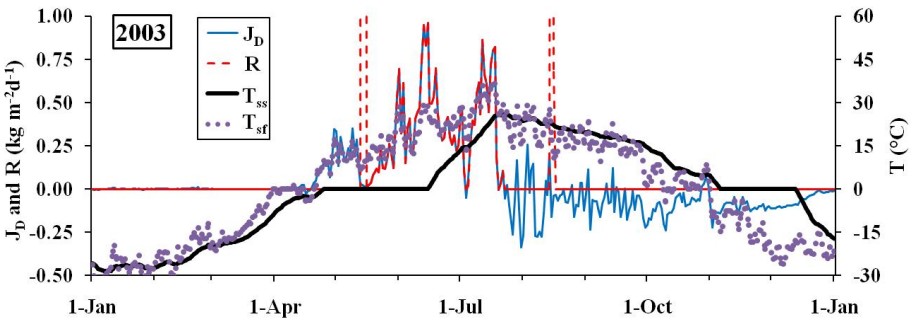


**Figure 4.** Zoom of recharge (red dashed line) and vapor diffusion (blue line) during 2003 (Temperatures are also shown).
Note that, except for the two heavy rainfall events of May and August, recharge during the warming period (after the subsoil
has started to thaw) is identical to the vapor diffusion flux.
Vapor diffusion is basically controlled by the diffusion coefficient (D), which is the only parameter that affects the vapor
diffusion flux significantly (Table 2). Reducing D leads obviously to a reduction of vapor diffusion. A similar effect results
from increasing the soil surface thickness, which results in an apparent reduction of the gradient. As less water is transported
downwards, evapotranspiration increases and recharge decreases.
As expected, evapotranspiration is the main sink of water. In fact, it is limited by water availability during the warm season,
being high only after rainfall events and during melting (Figure 3.b). The evapotranspiration is about 85% of rainfall (Table
2), which is similar to the results obtained by Ma et al., (2003) for the Selenge River basin, northern Mongolia. Note that the
evapotranspiration is very small during winter because low temperatures hinder vaporization. In fact, it is negative (i.e., ice
deposition) during January and February (Figure 5), when the soil is colder than the air. Sublimation only becomes relevant
in March. Cumulative sublimation (some 29 kg m$^{-2}$ year$^{-1}$ in 2003)is low compared to typical values of cold regions (see,
e.g., Zhou et al., 2014), but large compared to winter rainfall.
The whole cycle is driven by radiation, which follows the usual seasonal patterns, high during late spring and early summer
and low in winter, when net radiation may become negative, partly due to the high albedo of snow (Figure 3.d). The
radiation balance is highly sensitive to orientation of the slope (Table 3). Obviously, the south face receives more radiation
than the north face. But this is largely compensated by an increase in sensible heat flux. The sensible heat increases when
latent heat decreases. That is, heat is returned to the atmosphere either as latent heat when water is available for evaporation,
or as sensible heat when the soil is dry. According to the energy balance (Table 3), the sensible heat is higher than the latent
heat flux, which reflects the dry climate of region. As a result the effect of slope and orientation is smaller than we had
anticipated (Tables 2 and 3). The large increase in radiation of south facing slopes only results in a small increase in
evaporation and latent heat and a parallel reduction of infiltration and recharge decreases (Table 2),because there is little
water.





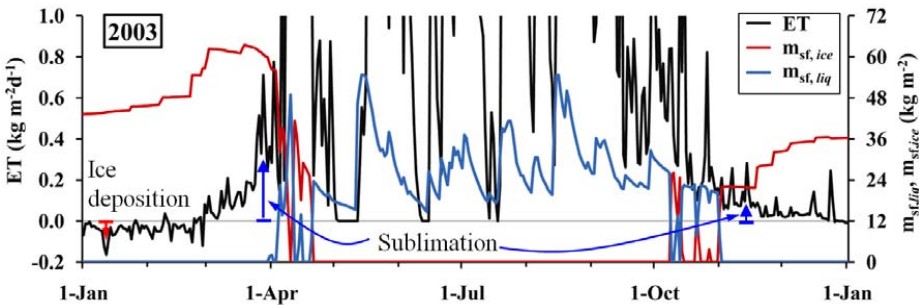

**Figure 5.** Zoom of ET (actually, water phase change processes) (black line) along with liquid (blue) and ice (red) water contents during 2003.

The dependence of the two balances on slope is non-monotonic, which points to the complexity of the system, even in the relatively simple model we are presenting here. Radiation is dramatically reduced in north facing slopes, which causes a reduction in ET, but the reduction is very small as discussed above, and not sufficient to cause an increase in infiltration. The reduction in ET is compensated by an increase in surface runoff and latent heat diffusion (vapor convection) downwards.

The non-monotonic dependence of vapor convection on slope also illustrates the robustness of vapor diffusion. It is slightly larger in south facing slopes than in horizontal land because surface temperatures are also larger. But it is also slightly larger in north facing slopes than in horizontal land because subsoil temperatures are lower.

**Table 2.** Average in 5 years of water fluxes: evapotranspiration (ET), infiltration (I), surface runoff (SR), vapor diffusion ($J_D$) and Recharge (R). Precipitation is 334 kg m$^{-2}$ year$^{-1}$)

| Water Bal. [kg m$^{-2}$ year$^{-1}$] | ET | I | SR | $J_D$ | R |
|---|---|---|---|---|---|
| Base model | 284.9 | 30.9 | 0.1 | 18.1 | 48.8 |
| South face | 290.8 | 23.8 | 0.0 | 19.5 | 43.3 |
| North face | 283.8 | 30.2 | 1.0 | 19.0 | 49.1 |
| $z_0$ (x 2) | 289.8 | 26.8 | 0.0 | 17.5 | 44.1 |
| $\varepsilon_s$ (0.94 → 0.99) | 280.6 | 35.5 | 1.0 | 16.9 | 52.2 |
| $K_{sat}$ (x 0.1) | 288.2 | 12.8 | 14.9 | 18.1 | 30.6 |
| $\theta^{wp}$ (x 2) | 252.9 | 66.0 | 2.1 | 13.1 | 79.1 |
| D (x 0.25) | 296.1 | 32.6 | 0.0 | 5.3 | 37.8 |
| $L_{sf}$ (x 2) | 318.5 | 4.2 | 0.0 | 11.4 | 15.6 |
| $L_{ss}$ (1.5 → 2.15) | 267.9 | 30.2 | 0.0 | 36.0 | 66.2 |





The limited availability of water causes most rainfall to evaporate. It also implies a low sensitivity of ET parameters to water and energy fluxes. For example, increasing roughness length ($z_0$) decreases the aerodynamic resistance ($r_a$), which leads to small increases in both latent and sensible heat fluxes and a parallel small decrease in infiltration and recharge. Similarly, increasing land surface emissivity from 0.94 to 0.99 reduces considerably net radiation as more longwave radiation is emitted, but is compensated by a decrease of sensible and, to a lesser extent, latent heat fluxes.

**Table 3.** Average in 5 years of energy fluxes: net radiation (Rn), Latent heat ($e_gET$), sensible heat (H), heat conduction (G) and vapor convection ($e_gJ_D$).

| Energy Bal. [MJ m$^{-2}$ year$^{-1}$] | **Rn** | **$e_gET$** | **H** | **G** | **$e_gJ_D$** |
|---|---|---|---|---|---|
| Base model | 2064.4 | 717.1 | 1345.6 | -45.8 | 45.9 |
| South face | 2610.7 | 732.3 | 1876.8 | -49.1 | 49.4 |
| North face | 1708.4 | 714.6 | 992.3 | -48.0 | 48.1 |
| $z_0$ (x 2) | 2181.4 | 729.5 | 1450.5 | -44.2 | 44.3 |
| $\varepsilon_s$ (0.94 → 0.99) | 1739.1 | 706.4 | 1030.9 | -42.9 | 42.9 |
| $K_{sat}$ (x 0.1) | 2067.2 | 725.4 | 1340.1 | -45.4 | 45.9 |
| $\theta^{wp}$ (x 2) | 2041.5 | 636.6 | 1402.1 | -34.4 | 33.1 |
| D (x 0.25) | 2072.4 | 745.5 | 1325.8 | -14.3 | 13.6 |
| $L_{sf}$ (x 2) | 1980.5 | 801.8 | 1177.9 | -28.6 | 29.0 |
| $L_{ss}$ (1.5 → 2.15) | 2032.0 | 675.2 | 1355.5 | -91.4 | 91.1 |

The most surprising energy balance terms are heat conduction (soil heat flux) and vapor convection (Figure 3.e). Conductive heat flux is usually considered seasonal, with yearly averages close to zero. Downward heat fluxes in summer are usually balanced by upward fluxes in winter (e.g. Alkhaier et al., 2012). In our case, even though the subsoil remains frozen for long (more than 7 months, compared to less than six the surface layer), there is a net flux upwards, to compensate the latent heat convection associated to vapor diffusion, which is downwards, as was discussed before. Therefore, it is not surprising that all factors that reduce the soil heat flux cause an increase in vapor convection, and vice versa.

The temperature oscillates more at the surface than at the subsoil layer (Figure 3.f). The differences of temperature of the layers are higher in summer. One can observe clearly the periods of melting and freezing of the subsoil layer with temperatures of 0°C. The annual average surface layer temperature was 0.8°C for 2003 and 2.0°C for 2004 while the air temperature was -3.6°C for 2003 and -2.4°C for 2004. For subsoil layer, -0.5°C and 0.7°C, in 2003 and in 2004 respectively.

An increase of the subsoil length ($L_{ss}$) leads to temperatures in the subsoil that oscillate less due to the increased heat storage capacity. This leads to larger temperature differences between surface and subsoil, which according to our model (equation 9, Figure 6) leads to larger vapor diffusion. As more water is transported downwards, evapotranspiration decreases and recharge increases.





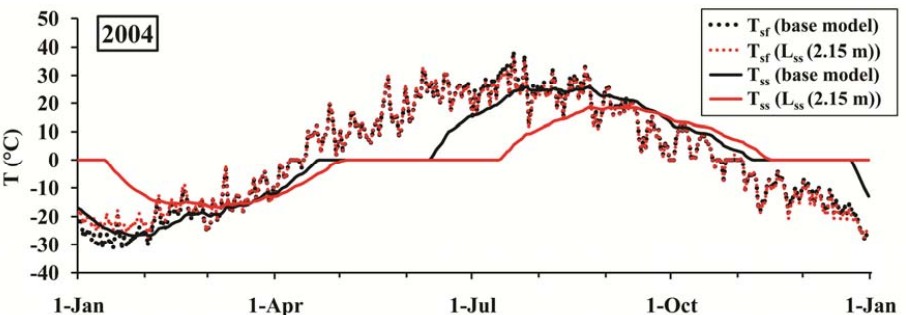

**Figure 6.** Daily evolution of surface and subsoil temperature in 2004.
**4. Discussion and Conclusions**
We have developed a water and energy balance model that contains two layers and attempts to represent all terms relevant
for simulating land surface hydrological processes, including all possible phase changes and, singularly, vapor diffusion. The
model has been applied using meteorological data from the Terelj station, northern Mongolia and typical soil properties of
the region. Results are consistent with local observations by others:
- Direct surface runoff is negligible and restricted to snowmelt periods.
- Liquid infiltration and subsequent recharge are restricted to a few heavy rainfall events. However, a sizable
recharge (about half of that from rainfall events) occurs continuously during late spring and early summer.
- Evapotranspiration is limited by water availability, as it accounts for 85% of rainfall. Sublimation is restricted to
late fall and spring, but it is also large, compared to winter snowfall. Ice deposition occurs most days during January
and February.
- Sensible heat is higher than latent heat flux, which reflects the dry climate of the region and low precipitation.
- The active layer remains frozen during the winter with periods of freezing or thawing of some three months, a
length of time that increases when the thickness of the active layer increases.
In summary, results are qualitatively consistent with observations. Notably, total runoff would be too small, compared to
observations, if vapor diffusion is reduced. The most singular result of the simulations is the relative importance of vapor
diffusion, which is downwards during spring and early summer, when temperature and, therefore, vapor pressure are higher
in the surface than in the subsoil. The upwards vapor diffusion flux is much smaller than the downward one, because vapor
pressure is a non-linear function of temperature. This downward flux transforms into recharge, which is continuous, although
fluctuating during that period.
In summary, the net downward vapor flux is relevant both in terms of water balance, accounting for a sizable portion of
recharge, and energy balance, causing a net upwards flux of heat. We conclude that land surface schemes should account for
vapor diffusion. We notice that, being a diffusive process, it may be included in such schemes at a moderate effort. Still,





further research is needed to ascertain the right values of diffusion coefficient to be used and the way of discretizing Fick's
Law, that is, the choice of length over which diffusion takes place in equation (9).
**Appendix A**
**Position of the sun**
For the calculation of the position of the sun and the zenith, it is convenient to define two unit vectors: $\mathbf{p}$, which is
orthogonal to the land surface and $\mathbf{s}$, which points to the sun (Figure A1). Their first component points eastwards, the second
northwards and the third upwards. Vector $\mathbf{p}$ can be calculated from the strike ($\sigma$) and dip ($\theta$) (see also figure A2):

$$\mathbf{p} = \begin{pmatrix} p_{east} \\ p_{north} \\ p_{up} \end{pmatrix} = \begin{pmatrix} \cos\sigma\sin\theta \\ -\sin\sigma\sin\theta \\ \cos\theta \end{pmatrix} \tag{A1}$$

Vector $\mathbf{s}$ depends on the time of the day, the sun declination ($\delta$), solar angle ($\omega$) and the latitude ($\varphi$). It can be calculated
according to Sproul (2007) as:

$$\mathbf{s} = \begin{pmatrix} s_{east} \\ s_{north} \\ s_{up} \end{pmatrix} = \begin{pmatrix} -\cos\delta\sin\omega \\ \sin\delta\sin\varphi - \cos\delta\sin\varphi\cos\omega \\ \cos\delta\cos\varphi\cos\omega + \sin\delta\sin\varphi \end{pmatrix} \tag{A2}$$

The product of both vectors ($\mathbf{p}^T\mathbf{s}$) equals the cosine of the angle between them. Note, that on a horizontal surface ($\theta = 0$), $\mathbf{p}^T$
is (0,0,1) and $\mathbf{p}^T\mathbf{s} = s_{up}$. At night $s_{up} < 0$, at daylight $s_{up} > 0$ and at sunrise and sunset $s_{up} = 0$. Furthermore, an inclined
surface is in the shade when $\mathbf{p}^T\mathbf{s} < 0$.
The sun declination ($\delta$) is the angle between the direction of the sun and the equator. It can be calculated by a yearly
sinusoidal function:

$$\delta = -\delta_{\max}\sin\left(2\pi\frac{t - t_s}{d_a}\right) \tag{A3}$$

where $\delta_{\max}$ is the maximum sun declination (0.4091 rad = 23.26°), t is time (s) , $t_s$ is time at September equinox,
approximately September 21st, (s) and $d_a$ is the duration of year (=365.241 days =3.15568×10^7 s).

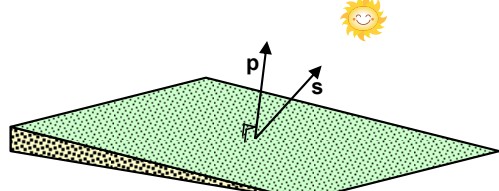

**Figure A1.** Illustration of vectors $\mathbf{p}$ and $\mathbf{s}$



The extraterrestrial solar radiation on a horizontal surface can be simplified by:

$$R_{S,et,hor} = S_0 f_e s_{up} = S_0 f_e (\cos\delta \cos\varphi \cos\omega + \sin\delta \sin\varphi) \qquad \text{If } s_{up} > 0$$

$$R_{S,et,hor} = 0 \qquad \text{otherwise} \qquad \text{(A4)}$$

where $S_0$ is the sun constant (1367 J m$^{-2}$ s$^{-1}$) and $f_e$ is the factor that corrects for the eccentricity of the earth´s orbit. It can be calculated from (Allen et al., 1998):

$$f_e = 1 + 0.033 \cos(2\pi \frac{t - t_{ph}}{d_a}) \qquad \text{(A5)}$$

where $t_{ph}$ is the time at perihelion, approximately January 3$^{rd}$, (s).

For the daily averaged extraterrestrial solar radiation on a horizontal surface, integrating equation (A4) between sunset and sunrise and dividing by the duration of a day ($d_d$) gives:

$$R_{S,et,hor} = \frac{S_0 f_e}{d_d} \int_{-\omega_{ss}}^{\omega_{ss}} s_{up} \, dt = \frac{S_0 f_e}{d_d} (\cos\delta \cos\varphi \sin\omega_{ss} + \omega_{ss} \sin\delta \sin\varphi) \qquad \text{(A6)}$$

where the sunset angle, $\omega_{ss}$, is the solar angle when $s_{up}$ equals 0:

$$\omega_{ss} = \cos^{-1}(\max(\min(-\tan\varphi \tan\delta, 1), -1)) \qquad \text{(A7)}$$

The min and max functions guarantee that $\omega_{ss} \in [0, \pi]$. Thus, equation (A6) also works for days when the sun doesn't set or rise (i.e., in polar regions).

**Correction for an inclined surface**

The strike ($\sigma$) of an inclined plane is the orientation of a horizontal line on this plane, expressed as an angle relative to the north in clockwise direction. The dip ($\theta$) is the maximum angle between a horizontal plane and the incline plane (figure A2).

The extraterrestrial solar radiation on an inclined surface is the solar radiation without taking into account the reduction of it by the atmosphere. It can be expressed as:

$$R_{S,et,inc} = S_0 f_e \max(\mathbf{p}^T \mathbf{s}, 0) \qquad \text{If } s_{up} > 0$$

$$R_{S,et,inc} = 0 \qquad \text{otherwise} \qquad \text{(A8)}$$

Note that for a horizontal surface ($\theta = 0 \Rightarrow \mathbf{p}^T \mathbf{s} = s_{up}$) equation (A8) reduces to (A4). For the daily averaged extraterrestrial solar radiation on an inclined surface, we have to integrate equation (A8):

$$R_{S,et,inc} = \frac{S_0 f_e}{d_d} \int_{-\omega_{ss}}^{\omega_{ss}} \max(\mathbf{p}^T \mathbf{s}, 0) dt \qquad \text{(A9)}$$





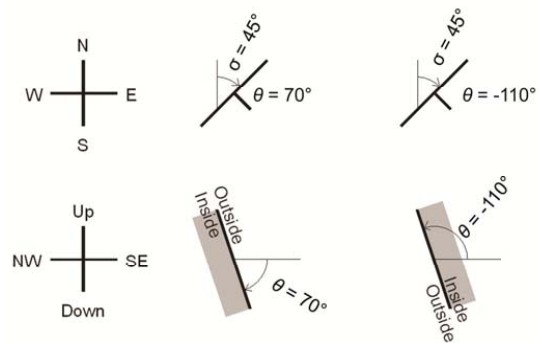

**403**

**404**    **Figure A2.** *Illustration of strike (σ) and dip (θ). Note that a dip between -0.5π and 0.5π (-90⁰ and 90⁰) refers to a plane with*

**405**    *its outside facing upwards. A dip between 1.5π and 0.5π (90⁰ and 270⁰) or between -1.5π and -0.5π refers to a plane with its*

**406**    *outside facing downwards.*

**407**    However, we have to take into account that the surface can be in the shade during part of the day, which complicates the

**408**    calculations. The integral of $\max(\mathbf{p}^T\mathbf{s}, 0)$ can be calculated by dividing it into different periods, when it is in the shade or

**409**    not:

$$\int_{-\omega_{ss}}^{\omega_{ss}} \max(\mathbf{p}^T\mathbf{s}, 0)\, dt = \max\left(\int_{-\omega_{ss}}^{\omega_1} \mathbf{p}^T\mathbf{s}dt, 0\right) + \max\left(\int_{\omega_1}^{\omega_2} \mathbf{p}^T\mathbf{s}dt, 0\right) + \max\left(\int_{\omega_2}^{\omega_{ss}} \mathbf{p}^T\mathbf{s}dt, 0\right)$$

$$\omega_1 = \min\left(\max(-\omega_{ss}, \omega_{io1}), \omega_{ss}\right)$$    (A10)

$$\omega_2 = \min\left(\max(-\omega_1, \omega_{io2}), \omega_{ss}\right)$$

**410**    where $\omega_{io1}$ and $\omega_{io2}$ are the two solar angles in a day when the inclined surface comes out of or into the shade, that is, when

**411**    $\mathbf{p}^T\mathbf{s} = 0$. They are calculated from:





$$\text{if} -1 \geq \frac{b}{a\sqrt{1 + b^2/a^2}} \geq 1 \quad \text{and} -1 \geq \frac{c}{a\sqrt{1 + b^2/a^2}} \geq 1 \quad \text{then}$$

$$\omega_{io1} = \min(\mathrm{mod}(\pi - \omega_b + \omega_c, 2\pi) - \pi, \mathrm{mod}(-\omega_b - \omega_c, 2\pi) - \pi)$$

$$\omega_{io2} = \max(\mathrm{mod}(\pi - \omega_b + \omega_c, 2\pi) - \pi, \mathrm{mod}(-\omega_b - \omega_c, 2\pi) - \pi)$$

$$\omega_b = \frac{b}{a\sqrt{1 + b^2/a^2}}$$

(A11)

$$\omega_c = \frac{c}{a\sqrt{1 + b^2/a^2}}$$

else (there is no solution for $\mathbf{p}^T\mathbf{s} = 0$)

$$\omega_{io1} = -\pi$$

$$\omega_{io2} = \pi$$

where mod is a function that returns the remainder of the first argument after it is divided by the second argument. This
guarantees that $\omega_1$ and $\omega_2 \in [-\pi, \pi]$. Moreover, the min and max functions guarantee that $\omega_{io1} \leq \omega_{io2}$. When there is no
solution for $\mathbf{p}^T\mathbf{s} = 0$, it means that during the whole day the inclined surface is facing the sun or not. The integral of $\mathbf{p}^T\mathbf{s}$ can
be calculated by using equations (A1) and (A2):

$$\int_{\omega_{ini}}^{\omega_{fin}} \mathbf{p}^T\mathbf{s}\, dt = \int_{\omega_{ini}}^{\omega_{fin}} (a \sin \omega + b \cos \omega - c)dt =$$

(A12)

$$= \frac{d_d}{2\pi}[-a(\cos \omega_{fin} - \cos \omega_{ini}) + b(\sin \omega_{fin} - \sin \omega_{ini}) - c(\omega_{fin} - \omega_{ini})]$$

with

$$a = -\cos \sigma \sin \theta \cos \delta$$

$$b = \sin \sigma \sin \theta \cos \delta \sin \varphi + \cos \theta \cos \delta \cos \varphi$$

(A13)

$$c = \sin \sigma \sin \theta \sin \delta \cos \varphi - \cos \theta \sin \delta \sin \varphi$$

*Sky view factor*





The sky view factor ($f_{sv}$), is the proportion of the sky above the inclined surface that is not blocked from view by the
surrounding horizontal plane (figure A3). It ranges from 0.5 for a vertical to 1 for a horizontal surface. In general one can use
the formula of Badescu (2002):

$$f_{sv} = \frac{\cos(2\theta) + 2}{4} \tag{A14}$$


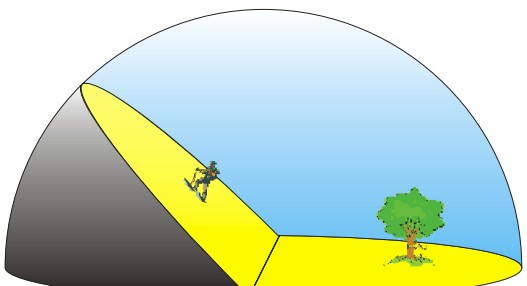

**Figure A3.** *Illustration of the sky view factor,* $f_{sv}$.

**Appendix B**
To test the validity of the discrete approximation of the diffusive vapor flux, we simulate the experiments of Gran et al
(2011a and b). Because of differences in the setting (Gran's experiments were performed in the laboratory), we had to do
several changes in the spreadsheet. These are highlighted in yellow in the spreadsheet, also available in the website together
with the original general use spreadsheet, and we discuss them here.
Meteorological data had to be changed. There is no rainfall, we assumed zero wind velocity and the net radiation was fixed
at 750 W m$^{-2}$, value reported by Gran et al., (2011a).
Since the experimental column loses heat through the column sides, we had to add a sensible heat sink of the form $\alpha(T_{env} -$
$T_{col})$ in both the subsoil (ss) and soil surface (sf) layers. When we adopted the value reported by Gran et al. (2011b) for $\alpha$,
the column cooled down too much. In examining her input files, we realized the value she reports in the paper is not correct.
The right value is far smaller.
The column thickness is 24 cm. So, we adopted a thickness of 2 cm for the top layer and 22 for the bottom.
As the thickness is small, capillary fluxes are relevant. We computed them as proportional to the difference in water content,
and adjusted the proportionality constant so as to fit observations (Figure B1)





The initial calculations (with large $\alpha$ for lateral heat exchange, see above) were unstable, so we had to reduce the time step
from 1 day to 1 hour, which was also a test of the internal consistency of the spreadsheet. As it turned out, results would
have also been stable with the final reduced $\alpha$, but we still kept the 1 h time step.
As shown in Figure B1, results are quite good (note that no diffusion or boundary layer parameters were touched from those
of the basic spreadsheet). So we left it at that, without trying to perform a formal calibration (an informal trial and error
calibration was effectively performed when adjusting the lateral heat exchange and the capillary flux).

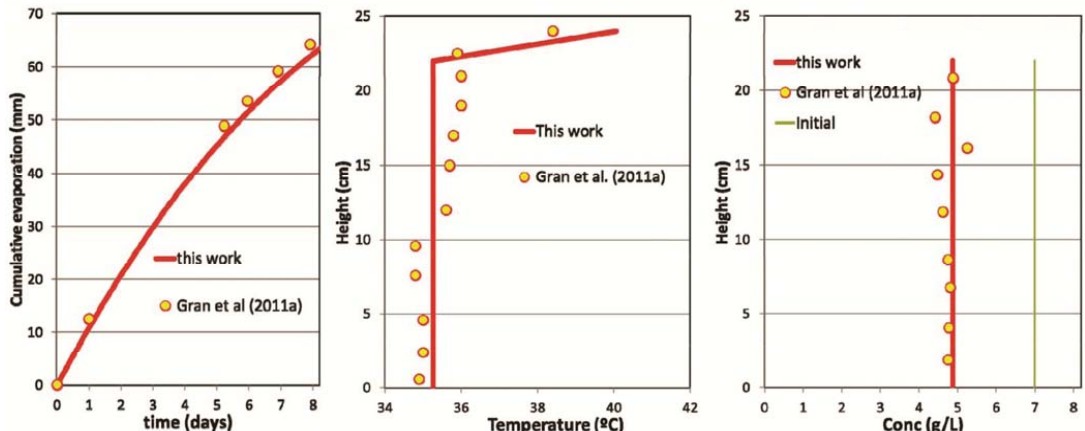


**Figure B1.** Time evolution of cumulative evaporation (left), spatial distribution of Temperature (center) and concentration
(right) at the end of the experiment of Gran et al. (2011a).

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
