# Peer review of "A surface model for water and energy balance in cold regions accounting for vapor diffusion"

_Hydrology and Earth System Sciences, 2017_

## Referee Comment (RC1) · Anonymous Referee #1 · 15 Oct 2017

This paper studies the role of vapor diffusion in the land energy and water balance cold and semi-arid regions. A new water and energy balance model has been developed that accounts for freezing and melting. In general, the topic is interesting for the land surface modelling community. However, there are several problems need to be addressed.

(1) Line 359: 'In summary, results are qualitatively consistent with observations.'. However, I did not find the evaluation of the model simulations with observations. It is important to comprehensively evaluate the new model in representing water and energy cycles (e.g., river discharge, soil/surface temperature, sensible/latent/ground heat

fluxes) over more than one site in different river basins.

(2) The equations are not precise enough. All of them should be re-checked. For example, in equation 16, Rnl should equal to RL,down - RL,up, but not RL,up + RL,down . Line 208, the description of $\lambda$ is puzzled. It is better to express by an equation.

(3) Lines 74-76, the inputs and outputs are mixed here. I suggest to describe the inputs/outputs in a more clearly way, not only for water, but also for energy balance.

(4) I do not see the advantages of the new model. The authors may want to emphasize it more clearly.

(5) In Equation 1 and Lines 96-99, how the interception loss (by canopy) is treated in the model ?

(6) Instead of using 'length' to explain Lss and Lsf , the 'depth' might be easier to understand. (e.g., Line 130 and other places)

(7) Equations 13-15, how the ml , mi and ms are determined in model ?

(8) How is the model initialized ? At what time step is the model run ?

---

## Author Comment (AC1) · 20 Oct 2017

Response to comments Anonymous Referee #1

This paper studies the role of vapor diffusion in the land energy and water balance cold and semi-arid regions. A new water and energy balance model has been developed that accounts for freezing and melting. In general, the topic is interesting for the land surface modelling community. However, there are several problems need to be addressed.

We respond to the comments in normal text, whereas the comments are numbers in

parentheses.

We thank the referee for his/her kind assessment of our work

(1) Line 359: 'In summary, results are qualitatively consistent with observations'. However, I did not find the evaluation of the model simulations with observations. It is important to comprehensively evaluate the new model in representing water and energy cycles (e.g., river discharge, soil/surface temperature, sensible/latent/ground heat fluxes) over more than one site in different river basins.

The referee is right in that we have not tested the model in a quantitative way for the Tuul River basin. There are two reasons for that. First, data are limited. Long records of output variables are available only for river discharge and snow depth. A proper modeling of the basin requires spatial discretization to acknowledge variability, which is relevant for both rainfall and snow depth. Therefore, a description of the model would be far too long. We are preparing another work comparing model results with observation data. Second, and most important, while our work was motivated by the Tuul River basin, the primary goal of this paper was to show the importance of vapor diffusion fluxes within the soil, which are ignored by most models. The validity of such concept can only be tested qualitatively. Thus we tested model validity by comparison to other studies in the region (Ma et al., 2003; Zhou et al., 2014). We also tested our model with laboratory experiment's data (Gran et al., 2011a) which is presented in appendix B.

(2) The equations are not precise enough. All of them should be re-checked. For example, in equation 16, Rnl should equal to RL,down - RL,up, but not RL,up + RL,down. Line 208, the description of $\lambda$ is puzzled. It is better to express by an equation.

We have revised the equations independently a few times and will do again prior to submitting the revised version of the paper. Yet, typos are a pest hard to eliminate. In fact, a typo in line 208 has made the description of thermal conductivity "puzzling", the reviewer is right. It should read:

"where $\lambda$ is the soil thermal conductivity ($\lambda=\lambda w\hat{~}phi * \lambda s\hat{~}(1\text{-}phi)$), where phi is the porosity, $\lambda s$ is the thermal conductivity of the solid particles, and $\lambda w$ is the thermal conductivity water, equal to that of ice, $\lambda i$, when water is frozen or liquid water, $\lambda l$, otherwise (Côté and Konrad, 2005)."

Regarding the sign of RL,up, our sign convention is that upward fluxes are negative, which facilitate handling terms where the flux can be positive or negative. We agree that it may be confusing to treat RL,up, which is always upward, as negative (see eq. 21), but it should be written this way for consistency. Still, acknowledging the potential for confusion, we will clarify it in the revised version.

(3) Lines 74-76, the inputs and outputs are mixed here. I suggest to describe the inputs/outputs in a more clearly way, not only for water, but also for energy balance.

For water balance, main input is the precipitation as rain or snow. For surface layer, outputs include the evapotranspiration (both ice deposition and sublimation), surface runoff and the infiltration into the subsoil while the recharge into aquifer for subsoil soil. Vapor diffusion occurs between the two layers.

For energy balance, main input is the net radiation. For surface layer, outputs consider latent heat, sensible heat and heat fluxes whereas conduction between the two layer and energy released due to phase changes.

(4) I do not see the advantages of the new model. The authors may want to emphasize it more clearly.

As mentioned above (see response to comment 1), but also in the paper, the main goal of our work is to show the importance vapor diffusion, which is ignored by most land surface models (e.g., SWAT). As such, our primary goal is not so much to develop a new model (although we consider it pretty complete, also in regards to radiation) as to argue that widely used existing models should incorporate diffusion. None of these integrated models simulate vapor diffusion in the soil. The goal of our work is to gain

insight into the hydrological processes in cold regions. We analyze the sensitivity of the model to the values of the parameters, which implicitly allows us to assess the relative importance of the various processes in the soil. It is from this analysis that we conclude that vapor diffusion is a relevant mechanism that should not be ignored. We will add a sentence at the end of the abstract and conclusions to emphasize this point.

(5) In Equation 1 and Lines 96-99, how the interception loss (by canopy) is treated in the model?

The canopy is important both for interception and for evapotranspiration. We do not model interception loss explicitly because, our goal is not so much flood prediction (for that we would have to integrate many other processes) as to assess the relevance of vapor diffusion, which is a slow process. Interception is usually evaporated within a few hours after rainfall and it is incorporated in the soil layer, which is usually sufficient for water resources assessment models. As for the role of the canopy in evapotranspiration, the surface resistance, rs, describes the resistance of vapor flow through stomata openings, total leaf area and evaporating soil surface (Allen et al., 1998). It shows in equation 6.

(6) Instead of using 'length' to explain Lss and Lsf, the 'depth' might be easier to understand. (e.g., Line 130 and other places)

We thank the reviewer for the suggestion, because length might suggest horizontal extent, but we are not sure. Depth would be the distance from the surface, which is not our case. Our model is simulated in the midpoint of the surface and the subsoil layers. Therefore, we propose denoting Lss and Lsf as thicknesses.

(7) Equations 13-15, how the ml, mi and ms are determined in model?

We thank the referee for this comment, as the explanation was confusing: ms is the mass of solid, and ms=(1-phi)*rho_s*L, where rho_s is solid phase density (2650 kg/m3), phi is porosity, and L is the layer thickness. The total mass of water m, which

is known from the mass balance, is equal to the sum of the liquid (l) and ice (i) masses (m = ml + mi). The total internal energy is the sum of that of solid, liquid water and ice (U = ms*cs*T + ml*cl*T + mi(ci*T - $\Lambda$melt)). At temperatures above 0°C all water is liquid (m = ml) and at lower temperature all water is ice (m = mi). Melting and freezing occur at 0°C, and the fraction of water frozen can be obtained from the internal energy, U (Eq. 14). The concept is illustrated in Figure 2. We will clarify the explanation in the revised version of the paper.

(8) How is the model initialized? At what time step is the model run? The model was initialized by running it twice, to ensure negligible storage variations that would blur the balances. The conditions at the end of the first run were adopted as initial conditions for the second and final run, which is the one reported in the paper. We solve the water and energy balance equations using a semi-implicit finite differences scheme with a time step of one day. For modeling the evaporation experiments (Appendix B), we adopted a time step of 1 hour.

References

Allen, R, G., Pereira, L, S., Raes, D. and Smith, M.: Crop evapotranspiration-Guidelines for computing crop water requirements-FAO Irrigation and drainage paper 56, FAO, Rome., 300(9), D05109, 1998.

Côté, J. and Konrad, J.M.: A generalized thermal conductivity model for soils and construction materials, Canadian Geotechnical Journal., 42(2), 443-458, 2005.

Gran, M., Carrera, J., Massana, J., Saaltink, M.W., Olivella, S., Ayora, C. and Lloret, A.: Dynamics of water vapor flux and water separation processes during evaporation from a salty dry soil. Journal of hydrology, 396(3), 215-220, 2011a.

Ma, X., Yasunari, T., Ohata, T., Natsagdorj, L., Davaa, G. and Oyunbaatar, D.: Hydrological regime analysis of the Selenge River basin, Mongolia, Hydrological Processes., 17(14), 2929-2945, 2003.

Zhou, J., Pomeroy, J. W., Zhang, W., Cheng, G., Wang, G., and Chen, C.: Simulating cold regions hydrological processes using a modular model in the west of China, Journal of Hydrology., 509, 13-24, 2014.

———————————————————

---

## Referee Comment (RC2) · F. Dominé (Referee) · 12 Jan 2018

The main purpose of this work is to test using mostly a modeling approach whether vapor diffusion in the soil of a cold and semi-arid region is a significant process worthy of being incorporated into integrated hydrological models. The authors model the water and energy budget of a soil in Mongolia and conclude that it is indeed an important process. They find that the temperature gradient between the cold ground and warm air in spring leads to water vapor condensation in the upper soil layer, which greatly modifies both the water and energy budgets. In particular, it is important for the thawing of the active layer.

The question addressed is truly worthwhile and has often been neglected not just for the soil but also for the snowpack water budget. (Sturm and Benson, 1997) studied water vapor exchanges between snow layers and between the soil and snow layers in a cold and fairly dry region of interior Alaska which may be fairly similar to that studies by the authors and indeed found that in winter soil moisture migrated upward and condensed in the snow, so that there was an overall water loss from the soil because of vapor diffusion in soil. (Domine et al., 2016) studied the evolution of the temperature and water content of the soil at 10 cm depth at a high Arctic site and also observed water loss over the course of winter, which they also ascribed to water vapor diffusion from the warm soil into the colder snow.

Now, in this study, the authors find the opposite. Their Figure 4 shows that the flux of water vapor between the soil and the surface is zero during the winter months (mid-December to April) but that from mid-April to mid-August the water flux is from the warmer surface (or atmosphere) to the colder soil. On a yearly basis, there is a net flux of water vapor into the soil, which is found to be critical for recharge and for active layer thawing.

It is clear that there is a discrepancy between both experimental studies cited and the model results of this study, and this is worrisome. In particular, the zero water vapor flux in winter in the model in troubling, because both experimental studies conclude to a very significant soil water vapor loss over that period, and this issue must be resolved before publication. The purpose of this review is to suggest reasons for this discrepancy, which may perhaps lead to useful model modifications by the authors. The model divides the soil into just 2 layers: the surface layer, which is 16 cm thick, and the subsoil which extends down to 150 cm. I did not find a description of how the snow layer was treated (besides its albedo) and I wonder whether the thermal impact of the snow was even treated in the model. In winter, the snow protects the ground from the winter cold air, with the result that the ground surface is much warmer then the snow surface. This is illustrated e.g. in (Domine et al., 2016) but also in countless other

papers, showing that a thin snow cover leads to a soil surface temperature warmer by at least 10°C than the air. If this effect is not taken into account, then the simulated soil temperature will be too cold, and therefore the water vapor flux will be greatly reduced. In any case, simulating a negligible soil drying over winter seems contrary to published work, and also to all my observations in cold regions, where I have always observed very dry soils at the end of winter.

The thickness of the layers may also cause errors. If my understanding is correct, the temperature of each layer is the average temperature of the layer, while in fact exchanges with the atmosphere will be dictated by the difference in temperature between the air near the surface and the soil very surface. I believe this may cause very large errors. After snowmelt, the surface can warm up very rapidly in the presence of solar radiation. Radiation absorption by the surface is even the reason why the air warms up, as heat is transferred from the hot surface to the colder air in the daytime, as the authors doubtless know. If the average temperature of a 16 cm layer is considered, then clearly this average temperature will be significantly colder than the surface temperature, because the soil is warming up in spring and summer. Calculating water vapor exchanges using that temperature can only lead to inadequate conclusions, and most likely to the wrong sign of the flux.

Other aspects of the model are surprising or arguably approximate. Using 0.6 for snow albedo (line 239) is extremely low. Perhaps it does apply to the actual site studied, but this would need qualification as snow albedo is almost always much greater (Gardner and Sharp, 2010), except when large amounts of vegetation protrude above the snow (Sturm et al., 2005; Loranty et al., 2011). An earlier statement (line 162) that albedo was determined from snow depth makes this all very confusing. For downwelling irradiance, why not use SBDART (Ricchiazzi et al., 1998)?

In summary, while the objective of this paper is interesting and laudable, I am very concerned that the model structure is not adequate (probably much too simple) to allows testing the objectives stated. I would recommend treating the thermal effects

of snow adequately (this may have been done, I just did not find the expected details) and using more soil layers with several thin layers near the surface. If water vapor exchanges are to be calculated reliably, the temperature of the top soil and of the air very near the surface must be calculated accurately. This probably requires 1 cm-thick soil layers near the surface. Finally, a convincing validation of the model would require measurements of the soil temperature and water content, preferably at several depths. Without such data, confidence in the model will remain very limited. Very major changes therefore seem required before publication.

References cited

Domine, F., Barrere, M., and Sarrazin, D.: Seasonal evolution of the effective thermal conductivity of the snow and the soil in high Arctic herb tundra at Bylot Island, Canada, The Cryosphere, 10, 2573-2588, 2016.

Gardner, A. S. and Sharp, M. J.: A review of snow and ice albedo and the development of a new physically based broadband albedo parameterization, J. Geophys. Res., 115, 2010.

Loranty, M. M., Goetz, S. J., and Beck, P. S. A.: Tundra vegetation effects on pan-Arctic albedo, Environ. Res. Lett., 6, 024014, 2011.

Ricchiazzi, P., Yang, S. R., Gautier, C., and Sowle, D.: SBDART: A research and teaching software tool for plane-parallell radiative transfer in the Earth's atmosphere, Bull. Am. Meteorol. Soc., 79, 2101-2114, 1998.

Sturm, M. and Benson, C. S.: Vapor transport, grain growth and depth-hoar development in the subarctic snow, J. Glaciol., 43, 42-59, 1997.

Sturm, M., Douglas, T., Racine, C., and Liston, G. E.: Changing snow and shrub conditions affect albedo with global implications, Journal of Geophysical Research-Biogeosciences, 110, G01004, 2005.

---

## Author Comment (AC2) · 22 Jan 2018

**Response to comments by Referee #2 Dr. Florent Dominé**

Dr. Florent Dominé raises a number of issues that question the validity of our paper. Before addressing them, we want to thank him for his thoroughness (it is clear that he has understood clearly our work), for the constructive tone of his comments, and for having introduced us to the literature on vapor diffusion as a mechanism for snow metamorphism. The latter, though not requested by Dr. Dominé, makes it clear that we need to address the snowpack water balance literature in the revised version of our paper. We will make a small summary of the topic in the introduction and we will compare our results to those of snow-pack researchers. It is important, however, to bear in mind that our contribution is oriented to water resources assessment (evaluation of surface and, especially, ground water runoff), as properly acknowledged by Dr. Dominé. The main novelty is that vapor diffusion is important not only for soil water and energy balances, but also for snow-pack diagenesis.

The three main issues raised by Dr. Dominé are (1) the apparent contradiction between our results and those of snow pack researchers; (2) the need for a more refined grid; and (3) the value of albedo. We respond below to these and other minor issues.

**(1) Winter vapor flux**

Based on our Figure 4 (reproduced here as Figure R.1), Dr. Dominé states that our winter vapor flux is zero, which contradicts results in the snow pack literature pointing that an upwards flux should be obtained. He is right in that the winter vapor flux should be upwards (since the deep soil is warmer than the surface, vapor pressure is larger at depth which leads to an upwards diffusive flux). However, upward vapor diffusion is highest during autumn and drops considerably during winter (mid December to April). This is also illustrated by, e.g., figure 12 of Dominé et al. (2016), (reproduced here as Figure R.2), which shows vapor fluxes in snowpack being highest in November. Actually, the magnitude and, especially, temporal evolution of vapor fluxes computed by Domine et al (2016) are comparable to ours. According to Domine et al (2016) vapor flux drops from 0.2 to 0.02 mg/m2/s (or from 0.017 to 0.0017 kg/m2/d, in our "hydrological" units) from Nov, 1[st], 2014 to Jan, 1[st], 2015. Ours starts at 0.2 kg/m2/d and drops more slowly at first, but faster towards the end of January. In both cases, the flux reverses by the end of April. Given the different settings (their study was performed at a much higher latitude than ours), we consider their work like an independent validation of ours.

This evolution of vapor fluxes can be explained easily by assuming the air in the soil (or snow) to be saturated and writing Fick's law as (compare eq. 9 of our article):

$$J_D = -\frac{MD}{RT}\frac{\partial p_{v,sat}}{\partial z} = -\frac{MD}{RT}\frac{\partial p_{v,sat}}{\partial T}\frac{\partial T}{\partial z} \qquad (R1)$$

The highest temperature gradients (*dT/dz*) are in autumn and drop in winter. Another effect is the fact that at low temperatures the saturated vapor pressure changes very little with temperature (that is, $dp_{v,sat}/dT$ is low). All these factors similarly acknowledged in the snowpack literature and in our work. Perhaps, the main difference stems from their emphasis on small scales (metamorphism within the snow cover), which is related to the spatial and temporal discretization, which is addressed below.

[Figure]

Figure R.1: Reproduction of Figure 4 of the Discussion paper. Note that the diffusive flux is negative.

[Figure]

Figure R.2. Vapor fluxes computed by Domine et al. (2016) at the snow pack

**(2) Spatial discretization**

Dr. Dominé questions our choice of two layers. The issue of discretization has got three sides: (1) whether it is sufficient for simulating heat conduction, (2) whether it is sufficient to simulate vapor fluxes; and (3) whether it is necessary to acknowledge the insulation effect of snow. We have analyzed the first two issues, but not the last one. We discuss all of them below. We realize the referee is only questioning the last issue, but we feel it is necessary to discuss the other three to properly respond without "ex-cathedra" reasoning.

**2.1 Are two layers (and 1-day time step) sufficient to simulate heat conduction?**

The issue largely depends on the goal. In fact, the appropriate grid size depends on the time increment. The rule of thumb we adopt is that $\Delta t$ should be of the order of

$C \cdot \Delta z^2 / \lambda$, where $C$ is thermal capacity and $\lambda$ is thermal conductivity. You do not loose accuracy by working with a smaller than required time step, but you do not gain much either.

To address this issue, we have simulated heat conduction through a 1.6 m thick soil with thermal conductivity of 0.7 W/m/K and thermal capacity of 3 MJ/m³/K. Temperature is prescribed at the surface a double sinusoidal, one with one day period and 20 ºC amplitude, to simulate daily temperature fluctuations, and one with one year period and 60 ºC amplitude, to simulate seasonal temperature fluctuations. This superposition is appropriate for mid-latitudes, though not above the polar circles (Domine's study was performed at a 70+ latitude, where the sun does not rise during winter).

We first simulate heat conduction with a 50 nodes grid ($\Delta z = 3.2$ cm) using time increments of 1d and 0.1 d. We then average computed temperatures for the top 5 elements (i.e., 0.16 m, our sf layer) and the bottom 45 elements (i.e., our ss layer). Results are shown in Figure R.3. It is obvious that the solution with 1 d intervals miss the impact of daily fluctuations (see zoom at the right of Fig. R.3) at the sf layer (results are identical for the ss layer). But results are also identical at the sf layer, when the comparison is made in terms of daily averages.

[Figure]

Figure R.3: Temperatures computed with a 50 nodes grid and time increments of 0.1 and 1 day for a half year and (right) zoom over 5 days. Tsf represents average over the top 16 cm, and Tss represents average over the remaining 144 cm. In terms of averages, results are identical for 0.1 and 1 d time increments, but the latter misses the impact of daily fluctuations.

We now compare these solutions to those obtained with a 2 layers (cell centered) solution, similar to the one we used in the HESSD paper (except that here, we are adopting the boundary temperature at the edge of the top cell, to isolate the impact of spatial and time discretization). Results are shown in Figure R.4, which makes apparent that (1) in terms of daily averages, surface layer temperatures computed with 50 nodes are virtually identical to those computed with 2 cells; and (2) daily fluctuations with the two layers' model are not accurate (both in terms of amplitude and time lag).

In summary, the adopted discretization is adequate for simulating daily averages of temperature in the shallow layer (top 16 cm of the soil), although it would not have been appropriate to simulate daily fluctuations, which was not a goal of our model.

[Figure]

Figure R.4: Temperatures computed with a 50 nodes grid averaged over the top 16 cm (with a 0.1 d time step Tsf_0.1, averaged over 1d, Tsf_day_avg , which is virtually identical to the one with a 1 d time step, Tsf_1d) and the two layers' model of our HESSD paper (with a 0.1 d, Tsf_2l_0.1, and 1 d, Tsf_2l_1, time steps) for a half year and (right) zoom over 5 days.

**2.2 Are two layers sufficient to simulate vapor fluxes?**

Addressing this question is very tricky because it would require performing a non-isothermal, multiphase flow model. Therefore, we verified the simulation of vapor diffusion by comparison with the laboratory experiments of Gran et al (2011). Results, which are shown in Appendix B of our paper, are excellent. This exercise is quite demanding, because the experiments of Gran et al. (2011) were performed under stiff conditions (very dry soil surface). The fact that our model performed well suggests that, indeed, the simulation of vapor fluxes is appropriate.

However, those experiments were performed in the laboratory and did not include daily fluctuations. Note that both Figures R.3 and R.4, display significant differences between extreme daily temperatures and average Tsf. Therefore, a valid question would be if daily fluctuations have a long term impact on vapor fluxes. Gran et al (2018) have addressed this question and conclude that that they do (while the average of daily temperature fluctuations is zero, the average of daily vapor flux fluctuations is not), but the effect is very small for water resources assessment, which is the goal of our work.

**2.3 Is it necessary to acknowledge the insulation effect of snow?**

The thermal conductivity of snow is low compared with that of typical soils. Therefore, snow can insulate more effectively than soil, depending on snow depth (Romeroy and Brun, 2001). The thermal conductivity of snow varies from 0.04 to 0.25 W m$^{-1}$ K$^{-1}$. The thermal conductivity of snow measured by Domine et al., 2016 in Canadian high Arctic that 0.04 W m$^{-1}$ K$^{-1}$ for thin snow depth and 0.28 W m$^{-1}$ K$^{-1}$ during winter for thick snow depth. In fact, vapor diffusion changes the thermal conductivity of snow (Domine et al., 2016). We have used an average 0.15 W m$^{-1}$ K$^{-1}$ of thermal conductivity of snow for our study.

To address the issue raised by the referee, we need to recall how resistances to flow are simulated. The fact that the surface temperature is different from the mean layer temperature is acknowledged by adding a soil surface resistance to the boundary condition (eq. 25). That is, heat exchange is assumed to consist of two resistances is series. The first represents the resistance to (sensible) heat exchange between the surface and the atmosphere

$$H = \frac{\rho_a c_a}{r_a}(T_{sup} - T_{air}) \tag{R2a}$$

The second resistance represents the resistance to heat conduction between the surface and soil (mean layer) temperature

$$H = \frac{\rho_a c_a}{r_{sf}}(T_{sf} - T_{sup}) \tag{R2b}$$

Combining these two equations to eliminate $T_{sup}$ yields:

$$H = \frac{\rho_a c_a}{r_a + r_{sf}}(T_{sf} - T_{air}) \tag{R3}$$

This is identical to our equation (25). The issue is that in the discussion version of the paper $r_{sf}$ was calculated (our Eq. 26) from the thermal conductivity of the soil, as

$$r_{sf} = \frac{0.5 L_{sf} \rho_a c_a}{\lambda} \tag{R4}$$

where $\lambda$ is the thermal conductivity of the soil. To acknowledge the insulating effect of snow, we would need to add:

$$r_{sf} = \rho_a c_a \left(\frac{0.5 L_{sf}}{\lambda} + \frac{L_{snow}}{\lambda_{snow}}\right) = r_{soil} + r_{snow} \tag{R5}$$

Once $T_{sf}$ has been computed, temperature at the snow surface can be obtained from Eqs. (R2a) and (R3) as:

$$T_{sup} = T_{air} + \frac{r_a}{r_a + r_{sf}}(T_{sf} - T_{air}) \tag{R5}$$

The temperature at the interface between soil and snow can be computed analogously. Therefore, the comparison is sensitive to the relative values of $r_a$, $r_{soil}$, and $r_{snow}$. In our case, $r_a$ is very sensitive to wind velocity (sensible heat exchanged is enhanced by turbulence and high wind, so that $r_a$ is reduced proportionally to wind speed, Eq. 5, with a 0.1 m/s threshold). But it is typically some 10 times greater than $r_{soil}$. As for $r_{snow}$, it depends on the type and thickness of snow. While the thermal conductivity of snow can be much smaller than that of the soil, its thermal capacity is also much smaller. The effect of variations in these two parameters on fluctuations within the medium was analyzed by Slooten et al., (2010) (they analyzed on the impact of variations of hydraulic conductivity and storativity on the hydraulic response to sea tides, but the problem is mathematically identical to the one we are discussing here). For $\lambda_{snow} = 0.15$ W m$^{-1}$ K$^{-1}$, and thicknesses of the order of 5-10 cm, $r_{snow}$ is much greater (some 3-6 times greater) than $r_{soil}$, but smaller than typical values of $r_a$. Therefore, one can expect a moderate effect snow isolation.

This is illustrated in Figure R.5, where $r_a$, $r_{soil}$, and $r_{snow}$ are (1.14, 0.11 and 0.33 J/m$^2$/K/d, respectively. It is clear that the effect of resistances is severe (so severe that daily temperature fluctuations are eliminated, not shown). As one might expect, if no air resistance is acknowledged, computed Tsf is significantly different when snow resistance is acknowledged (Tsf_snow). In fact, the winter difference is some 10 ºC,

which coincides with Dr. Domine's perception of the temperature difference across snow and he insulating effect of snow (the fact that the number is identical must be considered coincidental, given the differences between his work and ours). However, the role of snow insulation is much smaller when atmospheric resistance is acknowledged. The maximum difference due to snow insulation is some 2ºC (difference between blue and brown line in Figure R.5), but the difference is negligible at mid and late fall, when vapor diffusion fluxes are maxima.

[Figure]

Figure R.5: Temperatures computed with the two layers model without atmospheric or snow resistance (Tsf_2l), with only snow resistance (Tsf_snow), with only atmospheric resistance (Tsf_air), and with both snow and air resistances (Tsf_sn+air).

As a result, the impact of acknowledging snow resistance to heat flow is small (Figure R.6)

[Figure]

Figure R.6: Vapor fluxes computed with our model between the shallow (sf) and the deep soil (ss) layers with the original model (blue) and after including snow resistance to heat conduction. It can be seen that the difference is small.

The summary of this long discussion is that rather large thermal gradients are to be expected in response to temperature fluctuations (along the day when the sun rises!) or

in windy days, when atmospheric resistance is small. These gradients can produce significant fluctuations of vapor fluxes, which is consistent with the results cited by the referee (Sturm and Benson, 1997; Domine et al., 2016) and those of Gran et al. (2018). These fluctuations may cause irreversible changes in snow morphology through sublimation, melting and freezing (vapor flux is a very powerful energy transport mechanism), but will tend to average out, so that their impact of water resources assessment should be moderate.

**(3) Value of Albedo and downwelling irradiance**

Dr. Domine argues that our value of albedo (0.6) is too low. Indeed, the albedo is an important parameter for energy balances, but it is highly variable for snow. Values range from 0.7-0.9 for fresh snow to 0.2-0.4 for old snow or dirty snow (Hall and Martinec, 1985). Oke (1987, table 1.1) gives values of 0.4 for old snow and 0.9 for fresh snow. In general, snow albedos in northern Mongolia are low. Dashtseren et al., 2014 calculated an albedo of 0.5. The reason is, that in our study area snowfall is relatively low and wind speed high during winter and spring. This causes high sublimation in relation to winter snowfall. Wimmer et al. (2009) estimated a sublimation/snowfall ratio of 80% for northern Mongolia. This leads to thin snow packs (thicknesses of 3 to 6 cm were measured), which affects the albedo, because it is affected by the underlying surface and by vegetation protruding through the snowpack (Dang et al., 2015). In this work we avoided the above complexity and used a constant albedo of 0.6, which is low but may represent a realistic average value for northern Mongolia. The albedo depends on snow depth in the sense that this value is applied if the snow depth is larger than 0.

SBDART model was suggested for calculating downwelling irradiance, which is a software tool that computes plane-parallel radiative transfer in clear and cloudy conditions within the Earth's atmosphere and at the surface (Ricchiazzi et al., 1998). However, it is already a quite complicated model. For our purpose, we think the simple approach for calculating irradiance suffices. Actually in hydrology similar simple methods are common (see, e.g., Allen et al., 1998, which is a standard method).

**(4) A final note**

Dr. Dominé concludes "while the objective of this paper is interesting and laudable, I am very concerned that the model structure is not adequate (probably much too simple) to allow testing the objectives stated". It is true that the model is very simplified, but still a lot more complex than the existing land surface schemes used. Models always have to simplify the reality. The required complexity of the model depends on the goal of the model, available data, and model scale (spatial and temporal).

The result from our dwelling into the fascinating snow processes literature is that (1) vapor diffusion is important also for snow metamorphism, and (2) at the short term, small scale, diffusion fluxes are strongly controlled by surface temperature fluctuations, and the thermal insulation provided by snow is critical, but (3) we conclude that for daily averages over greater thicknesses, the effect is much smaller.

In the revised version of the paper, we will address these issues both in the introduction and in the discussion of results and we will include the thermal resistance (insulating effect) of snow. We will also expand the discussion of snow albedo and irradiance.

**References**

Dang, C., Brandt, R. E., & Warren, S. G. (2015). Parameterizations for narrowband and broadband albedo of pure snow and snow containing mineral dust and black carbon. *Journal of Geophysical Research: Atmospheres*, *120*(11), 5446-5468.

Dashtseren, A., Ishikawa, M., Iijima, Y., & Jambaljav, Y. (2014). Temperature Regimes of the Active Layer and Seasonally Frozen Ground under a Forest Steppe Mosaic, Mongolia. *Permafrost and Periglacial Processes*, *25*(4), 295-306.

Domine, F., M. Barrere, D. Sarrazin, (2016). Seasonal evolution of the effective thermal conductivity of the snow and the soil in high Arctic herb tundra at Bylot Island, Canada. *The Cryosphere*, 10(6), 2573.

Gran, M., Carrera, J., Massana, J., Saaltink, M. W., Olivella, S., Ayora, C., & Lloret, A. (2011). Dynamics of water vapor flux and water separation processes during evaporation from a salty dry soil. *Journal of hydrology*, 396(3), 215-220.

Hall, D.K. and Martinec, J. (1985). Remote sensing of ice and snow. Chapman and Hall, New York, 189 pp.

Meritxell Gran M., J. Carrera and MW Saaltink (2018) Effect of thermal gradients and rainfall on vapor diffusion in dry soils. Submitted

Oke, T.R.: Boundary Layer Climates. 2nd edition. Halsted, New York, 1987.

Pomeroy, J. W., and Brun, E. (2001). Physical properties of snow. *Snow ecology: An interdisciplinary examination of snow-covered ecosystems*, 45-126.

Ricchiazzi, P., Yang, S. R., Gautier, C., and Sowle, D. (1998). SBDART: A research and teaching software tool for plane-parallell radiative transfer in the Earth's atmosphere, Bull. *Am. Meteorol. Soc.*, 79, 2101-2114.

Slooten, LJ, J. Carrera, E. Castro, D. Fernandez-Garcia (2010) A sensitivity analysis of tide-induced head fluctuations in coastal aquifers. *Journal of hydrology*, 393(3), 370-380.

Sturm, M. and CS Benson (1997) Vapor transport, grain growth and depth-hoar development in the subarctic snow, *J. Glaciol.*, 43, 42–59.

Wimmer, F., Schlaffer, S., Aus der Beek, T., & Menzel, L. (2009). Distributed modelling of climate change impacts on snow sublimation in Northern Mongolia. *Advances in Geosciences*, *21*, 117.